# Revised International Staging System (R-ISS) stage-dependent analysis uncovers oncogenes and potential immunotherapeutic targets in multiple myeloma (MM)

Ling Zhong[1,2,3†], Peng Hao[4†], Qian Zhang[1,2†], Tao Jiang[5†], Huan Li[1], Jialing Xiao[1], Chenglong Li[5], Lan Luo[6], Chunbao Xie[2], Jiang Hu[4], Liang Wang[1], Yuping Liu[1], Yi Shi[1,2], Wei Zhang[4]*, Bo Gong[1,2]*

[1]Department of Health Management, Sichuan Provincial People's Hospital, University of Electronic Science and Technology of China, Chengdu, China; [2]The Key Laboratory for Human Disease Gene Study of Sichuan Province and Institute of Laboratory Medicine, Sichuan Provincial People's Hospital, University of Electronic Science and Technology of China, Chengdu, China; [3]Natural Products Research Center, Institute of Chengdu Biology, Sichuan Translational Medicine Hospital, Chinese Academy of Sciences, Chengdu, China; [4]Department of Orthopaedics, Sichuan Provincial People's Hospital, University of Electronic Science and Technology of China, Chengdu, China; [5]Department of hematology, Sichuan Provincial People's Hospital, University of Electronic Science and Technology of China, Chengdu, China; [6]Chengdu University of Traditional Chinese Medicine, Chengdu, China

*For correspondence:
zhangwspine@163.com (WZ);
gongbo@med.uestc.edu.cn (BG)

†These authors contributed equally to this work

Competing interest: The authors declare that no competing interests exist.

**Abstract** Multiple myeloma (MM) accounts for ~10% of all haematologic malignancies. Little is known about high intratumour heterogeneities in patients stratified by the Revised International Staging System (R-ISS). Herein, we constructed a single-cell transcriptome atlas to compare differential expression patterns among stages. We found that a novel cytotoxic plasma cell (PC) population exhibited with NKG7 positive was obviously enriched in stage II patients. Additionally, a malignant PC population with significantly elevated expression of MKI67 and PCNA was associated with unfavourable prognosis and Epstein-Barr virus (EBV) infection in our collected samples. Moreover, ribonucleotide reductase regulatory subunit M2 (RRM2) was found and verified to promote proliferation of MM cell lines, suggesting RRM2 may serve as a detrimental marker in MM. The percentages of CD8+ T cells and NKT cells decreased along with R-ISS stages, reflecting the plasticity of the tumour immune microenvironment. Importantly, their crosstalks with myeloid cells and PC identified several potential immunotargets such as SIRPA-CD47 and CD74-MIF, respectively. Collectively, this study provided an R-ISS-related single-cell MM atlas and revealed the clinical significance of novel PC clusters, as well as potential immunotargets in MM progression.

## Editor's evaluation

The work constructed a single cell transcriptome atlas of bone marrow for R-ISS-staged multiple myeloma patients. It identified novel plasma cell clusters with clinical significance. The finding can help to develop more potential immunotargets for multiple myeloma.

**eLife digest** Multiple myeloma is a type of bone cancer. It affects the immune cells that make antibodies, known as plasma cells. These immune cells live in the bone marrow. As with many types of cancer, the chance of survival is highest when multiple myeloma is diagnosed early. It has three stages, labelled I, II, and III. People with stage I or II disease have better outcomes than those with stage III, but the exact reasons are unclear.

Bone marrow contains lots of different types of cells, which can affect the growth of a tumour. These include cancer-targeting cells, called killer T-cells, and cancer-supporting cells called myeloid cells. Understanding these cells and how they interact could shed light on the different stages of multiple myeloma. One way to do this is to use single cell sequencing, which looks at the genes in use inside each cell at any one time.

Zhong, Hao, Zhang, Jiang et al. examined the bone marrow of two healthy donors and nine people with different stages of multiple myeloma. This revealed two new groups of plasma cells. One group, highest in stage II patients, was protective, with the potential to kill cancer cells. The other, highest in people with more aggressive disease, was harmful, with the potential to divide rapidly. The sequencing also identified molecules that might be useful drug targets for the future. These included a gene that drove growth in the dangerous plasma cells, and several that might help tumours escape from the immune system.

It is becoming increasingly clear that the environment around a tumour has a huge role to play in its progression. Understanding how this environment changes over time could aid in the development of more targeted treatments. The next step is to find out more about the molecules identified here.

## Introduction

Multiple myeloma (MM) is characterized by uncontrolled proliferation of monoclonal plasma cells (PCs) and accounts for approximately 10% of all haematologic malignancies (*Rajkumar, 2020*). The Revised International Staging System (R-ISS) was developed to stratify MM patients into stages I, II, and III (*Palumbo et al., 2015*) with distinct outcomes (*Kastritis et al., 2017*) and treatment response (*Cho et al., 2017*), with or without the assistance of other clinical parameters (*Galieni et al., 2021*; *Jung et al., 2019*). However, the single-cell gene expression signatures in MM R-ISS stages remain to be explored. Epstein-Barr virus (EBV), the first human tumour virus (*Young et al., 2016*), causes Burkitt, Hodgkin, and post-transplant B cell lymphomas (*Wang et al., 2019*), and is associated with poor prognosis and clinical characteristics of R-ISS III in MM patients (*Xia et al., 2019*), although further studies are needed to uncover the mechanism. In the bone marrow (BM) microenvironment, the interplay between neoplastic cells and immune microenvironment cell is involved in MM progression and drug response (*Sperling and Anderson, 2021*), and single-cell level ligand-receptor interactions remain unclear. Myeloid cells (such as neutrophils) foster cancer-promoting inflammation, and natural killer (NK) cells and T lymphocytes mediate protective antitumour responses (*Guillerey et al., 2016*). In spite of dramatic advances in immunomodulatory drugs (*Tremblay-LeMay et al., 2018*; *Minnie and Hill, 2020*; *Nakamura et al., 2020*), monoclonal antibodies (*Hofmeister and Lonial, 2016*), proteasome inhibitors (*Ettari et al., 2018*), and histone deacetylase inhibitors (*Yee et al., 2016*), cases of MM still remain largely incurable (*Dimopoulos et al., 2015*).

Single-cell sequencing (scRNA-seq) offers an unprecedented opportunity to study the heterogeneity of PCs and immune microenvironments in cancer. Focusing on PCs, intratumour heterogeneity (*Ledergor et al., 2018*), genome evolution (*Lohr et al., 2016*), and transcriptome expression signatures (*Jang et al., 2019*), resistance pathways and therapeutic targets in relapsed MM (*Cohen et al., 2021*) were revealed. Meanwhile, compromised microenvironment immune cells (*Zavidij et al., 2020*) and transcriptional alterations (*Ryu et al., 2020*) in MM precursor stages and extramedullary progression were recently uncovered. Nevertheless, little is known about malignant PC and immune cell gene expression signatures with respect to the R-ISS stage and their role in EBV-infected MM at the single-cell level.

In this study, we adopted single-cell transcriptome sequencing to investigate the gene expression profiles in the normal and R-ISS stage I, II, and III groups. First, we examined the heterogeneity of PCs and validated the function and clinical significance of two PC populations. The function and clinical

significance of proliferating PCs and the hub gene ribonucleotide reductase regulatory subunit M2 (RRM2) were validated in other cohorts, cell lines, and collected samples. Next, gene expression modules underlying two T cell clusters with decreased proportions along with MM R-ISS stage were investigated. Finally, cell-cell communication was analysed to interpret the tumour cell-cytotoxic T cell and cytotoxic T cell-myeloid cell interactions in MM. Collectively, the results of this study provided a R-ISS-related single-cell MM atlas and revealed the clinical significance of two PC clusters, as well as potential immunotargets in MM progression.

## Results

### Single-cell transcriptome atlas of R-ISS-staged MM

To explore the intratumoural heterogeneity of R-ISS stage-classified MM, 11 BM samples from 9 MM patients (2 R-ISS I, 2 R-ISS II, and 5 R-ISS III stage) and 2 healthy donors (healthy control) were subjected to single-cell suspension preparation and transcriptome sequencing (*Figure 1A*). After low-quality cell filtering, removing double cell and batch effect (*Figure 1—figure supplement 1*), 101,432 single-cell expression matrices were acquired. Subsequent dimensional reduction generated 24 clusters (*Figure 1B*), with most clusters present in all four groups (*Figure 1C* and *Figure 1—figure supplement 1*) for subsequent analysis. Then, based on the cell type markers (*Table 1*Table 2), six general cell types were identified, consisting of PCs, B cells, myeloid cells, CD4+ T cells, CD8+ T cells, and immature red cells (*Figure 1D, E*). The proportions of 24 cell clusters in all groups are shown in *Figure 1F*.

### Functional identification of a NKG7+ PC population in MM

Intraclonal heterogeneity in PCs is emerging as a vital modulator in MM progression (*Dutta et al., 2019*), drug sensitivity, and therapeutic response (*Anguiano et al., 2009*). As is shown in *Figure 1D*, six clusters (4, 6, 13, 14, 18, and 22) specifically expressing high levels of CD138 (*Figure 2—figure supplement 1*) were classified into PCs. The PCs also displayed obvious positive for CD38 expression. Therefore, the identified PCs can be labeled as 'CD138+CD38+' cells in this study. Then, specific markers in each PC cluster were calculated. All six clusters showed abundant expression of ribosomal proteins such as RPS2,which corresponds to the antibody-producing function of PCs. The representative markers of each cluster were shown in *Figure 2A*, as LETM1 and NES in plasma cell cluster (PCC) 4, MKI67 and TOP2A in PCC6, LAMP5 and SERPINI1 in PCC13, CCND1 and XIST in PCC14, and IGLC2 in PCC22. A plasma cluster (PCC18) showing higher expression of the cytotoxic gene NKG7 (*Ng et al., 2020*; *Li et al., 2022*) caught our attention, and was proposed as 'cytotoxic PCs'. Coincidentally, Cupi et al. reported that PCs producing granzyme B (GZMB) showed cancer cell-targeting cytotoxic activities (*Cupi et al., 2014*), but studies of cytotoxic PCs in MM remain limited. The existence of cytotoxic PCs prompted us to investigate their existence and clinical relevance in MM. The average proportion of PC18 (in all PCs) was 12.12% (range 0–37.01%; *Figure 2B*, left panel), and no significant difference was observed between the four groups (*Figure 2B*, right panel). In another MM single-cell dataset focusing on PC heterogeneity of symptomatic and asymptomatic myeloma (dataset GSE117156) (*Ledergor et al., 2018*), one cluster, C21, exclusively expressing NKG7 corresponded to PC18 in our dataset (*Figure 2C, D*). In GSE117156 of all 42 samples, the cell proportion varied from 0% to 30.95% of all PCs, with an average percentage of 4.28% (*Figure 2E*). Next, immunofluorescence confirmed the expression of NKG7 in cytoplasm of PCs (CD138 positive) from patients with MM (*Figure 2F*). Finally, 20 MM patients (stage I: 3 patients, stage II: 10 patients, and stage III: 7 patients) were enrolled for multiparameter flow cytometric (MFC) analysis. The results showed that the percentage of NKG7+ PCs displayed obvious diversities among stage I, II, and III groups (*Figure 2G* and *Figure 2—figure supplement 1*). The average percentage of NKG7[+] population was 2.73% in stage I, 8.89% in stage II, and 0.58% in stage III (*Figure 2G* and *Figure 2—figure supplement 2*). In summary, we characterized an NKG7+ PC population (PC18), which may provide a novel perspective for the cytotherapy of MM.

### Clinical significance of a malignant PC population with high proliferation potential

Abnormal copy number variations (CNV) are a common feature of MM, which usually interferes with cell cycle checkpoints to induce accelerated proliferation (*Neuse et al., 2020*). To characterize

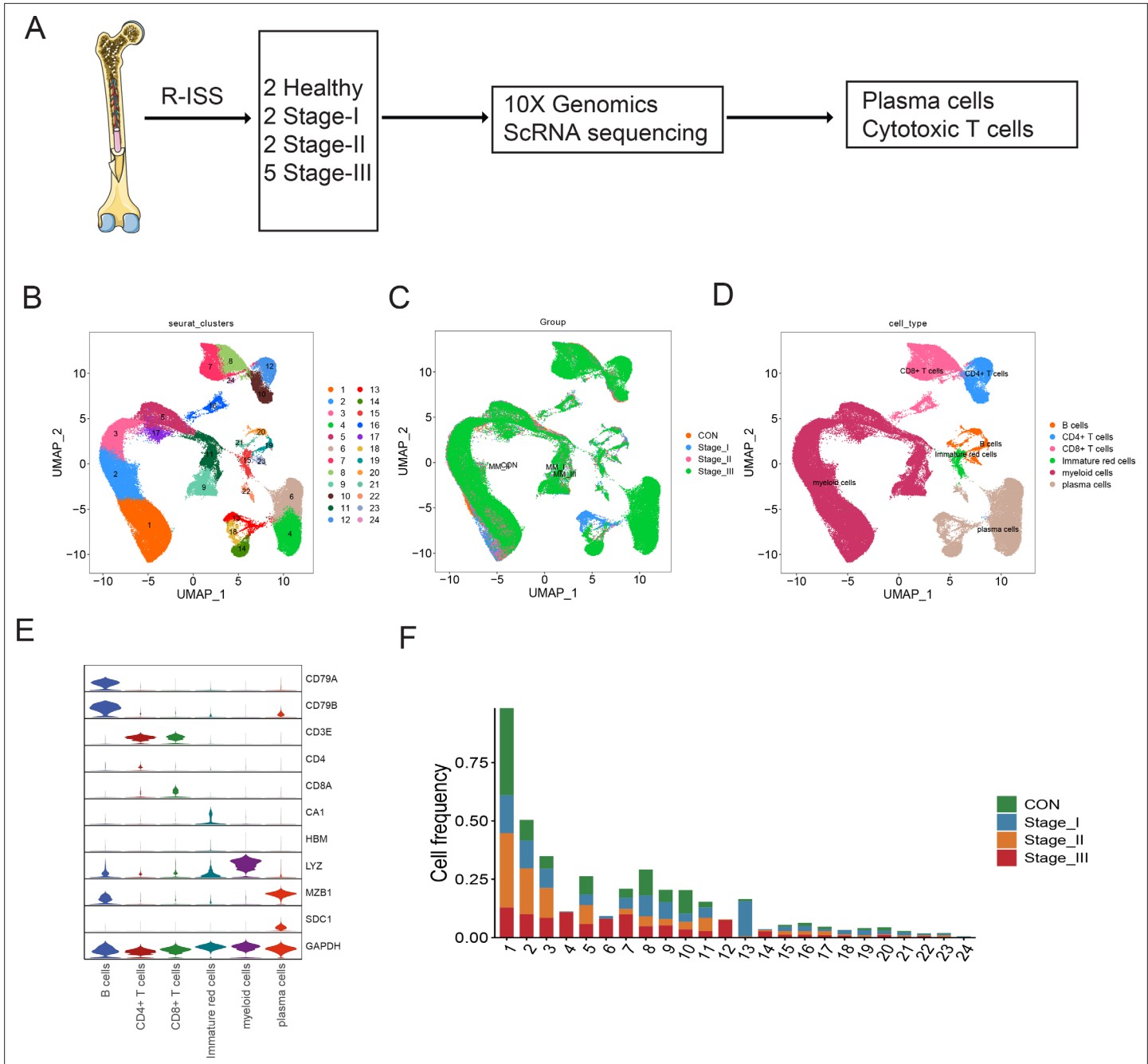

**Figure 1.** Single-cell transcriptome atlas of multiple myeloma (MM) with Revised International Staging System (R-ISS) staging. (**A**) Schematic illustration of workflow in this study. (**B**) Dimension reduction of cells, and 21 clusters were acquired, shown with UMAP. (**C**) UMAP showing the distribution of sample groups of normal, R-ISS I–III. Based on expression signature of canonical markers, six general cell types were identified. UMAP of cell type were shown (**D**), and violin plot of cell type markers in (**E**). (**F**) Proportion of cell clusters in normal and MM R-ISS groups.

The online version of this article includes the following figure supplement(s) for figure 1:

**Figure supplement 1.** UMAP of before and after batch-effect removing, and stage groups in multiple myeloma (MM).

malignant PCs and related oncogenes in MM, we first conducted inferCNV to delineate the CNV signals of all plasma clusters, especially in the five MM-III samples (*Figure 3—figure supplement 1*). The discrepancies exist in the five MM-III samples. Cytogenetic CNV signal between samples 4 and 5 are more similar to other samples. And MM-III-3 exhibited obvious amplication of chromosome 1 (*Figure 3—figure supplement 1*). As shown in *Figure 3A*, strong CNV signals were observed in

**Table 1.** Cell markers used for cell type identification.

| Cell type | Clusters | Cell markers |
|---|---|---|
| Plasma cells | 4,6,13,14,18,22 | MZB1,SDC1 |
| B cells | 19,20,21,23 | CD79A,CD79B |
| CD4+T cells | 10,12 | CD3D,CD3E,CD4 |
| CD8+T cells | 7,8,16,24 | CD3D,CD3E,CD8A |
| Myeloid cells | 1,2,3,5,9,11,17 | LYZ, S100A8 |
| Immature red cells | 15 | CA1, HBM |

plasma cell cluster 4 (PCC4) and 6 (PCC6), indicating their malignant features as tumour cells. Consistent with bulk genomic sequencing studies, signals were mainly located on chromosomes 1, 8 (gain/amplification) and 2, 3, and 21 (deletion) (*Samur et al., 2020*; *Manier et al., 2017*). Next, we analysed the cell cycle of six PC clusters, and distinguished them from other clusters, PCs in cluster 6 (PCC6) were presumably enriched in G2/M stage (*Figure 3B*). Consistently, cell proliferation-related oncogenic markers MKI67, TOP2A, and CDK1 also showed similar expression patterns over the whole cell cycle, suggesting a high proliferation potential of PCC6 in MM (*Figure 3B*). Similar to PCC6 in our dataset, the expression of MKI67, TOP2A, and CDK1 of cluster 11 in PCs of GSE117156 supported its very existence in MM (*Figure 3C*). Next, GSEA was conducted to investigate the signalling transduction in the two malignant clusters PCC4 and PCC6 (other PCCs in *Figure 3—figure supplement 2*). As shown in *Figure 3D*, oxidative phosphorylation and epithelial-mesenchymal transition were the most enriched hallmark pathways in PCC4 and PCC6 (*Carrasco et al., 2007*; *Hideshima et al., 2017*; *Rizq et al., 2017*; *Azab et al., 2012*). Finally, we applied PRECOG (*Gentles et al., 2015*) to delineate the cluster-gene expression specificity and the prognostic score in MM GSE6477 metadata. The results showed that most genes with high specificity in PCC6 also showed elevated prognostic z-scores, such as GGH (z-score=6.1), GINS2 (z-score=4.5), PRKDC (z-score=6.1), MCM3 (z-score=4.4), SHCBP1 (z-score=4.7), CHEK1 (z-score=3.9), and PHF19 (z-score=5.5) (*Figure 3E*). Consequently, we focused on PCC6, a malignant PC population with high proliferation potential and unfavourable prognostic significance, and investigated its clinical relevance and potential therapeutic targets in MM.

Next, we discovered 75 significantly up-regulated genes in the MM (UGM) dataset of GSE6477 compared with normal samples (*Figure 4A, B*). The specific genes (cluster specificity >0.6) in PCC6 were compared with 75 UGMs, and 20 genes were identified (*Figure 4C*). The expression of 20 UGMs is shown with the R-ISS stage in *Figure 4D*. We calculated a 20-gene signature score and analysed the relevance of the score with respect to clinical parameters (*Figure 4E*). Next, the prognostic significance of these 20 genes was analyzed. Four genes showed good performance as unfavourable markers, with hazard ratio (HR) for histidine triad nucleotide binding protein 1 (HINT1) of 1.9 (95% CI=1.2–2.9, p-value = 0.005496), translocase of inner mitochondrial membrane 13 (TIMM13) of 1.6 (95% CI=1–2.5, p-value = 0.03971), sigma non-opioid intracellular receptor 1 (SIGMAR1) of 1.6 (95% CI=1–2.5, p-value = 0.0493) and ribonucleotide reductase regulatory subunit (RRM2) of 2.3 (95% CI=1.4–3.6, p-value = 0.000402) (*Figure 4F*). As RRM2 showed the highest HR, it was chosen for subsequent assays. The expression of RRM2 was examined in MM patients and MM cell lines (MM.1S and U266). Consistently, significantly increased expression of RRM2 was observed in both clinical MM R-ISS III samples (*Figure 4G*) and both MM cell lines (MM.1S and U266) (*Figure 4H*). Next, the functions of RRM2 and HINT1 in the MM cell line U266 were studied. RRM2 and HINT1 silencing reduced the proliferation of U266 cells, respectively (*Figure 4I*). Next, the functions of RRM2 in the MM cell line U266 were studied. RRM2 silencing reduced the proliferation of U266 cells (*Figure 4I*), and induced cell cycle G1 phase arrest (*Figure 4J*). Moreover, elevated expression and unfavourable prognostic performance of HINT1 was found in MM, and further cellular assays shows that HINT1 silencing induced cell cycle arrest (*Figure 4—figure supplement 1*), implying its oncogenic role in MM, compared with its tumour-suppressing role in solid cancers. The above results shows that RRM2 specifically unregulated in proliferating PC serve as an oncogene in MM.

Finally, we identified the differentially expressed genes (DEGs) with fold change ≥2 or ≤0.5 and adjusted p-value <0.05 in PCC6 by comparing R-ISS stages I and III. A total of 351 DEGs were

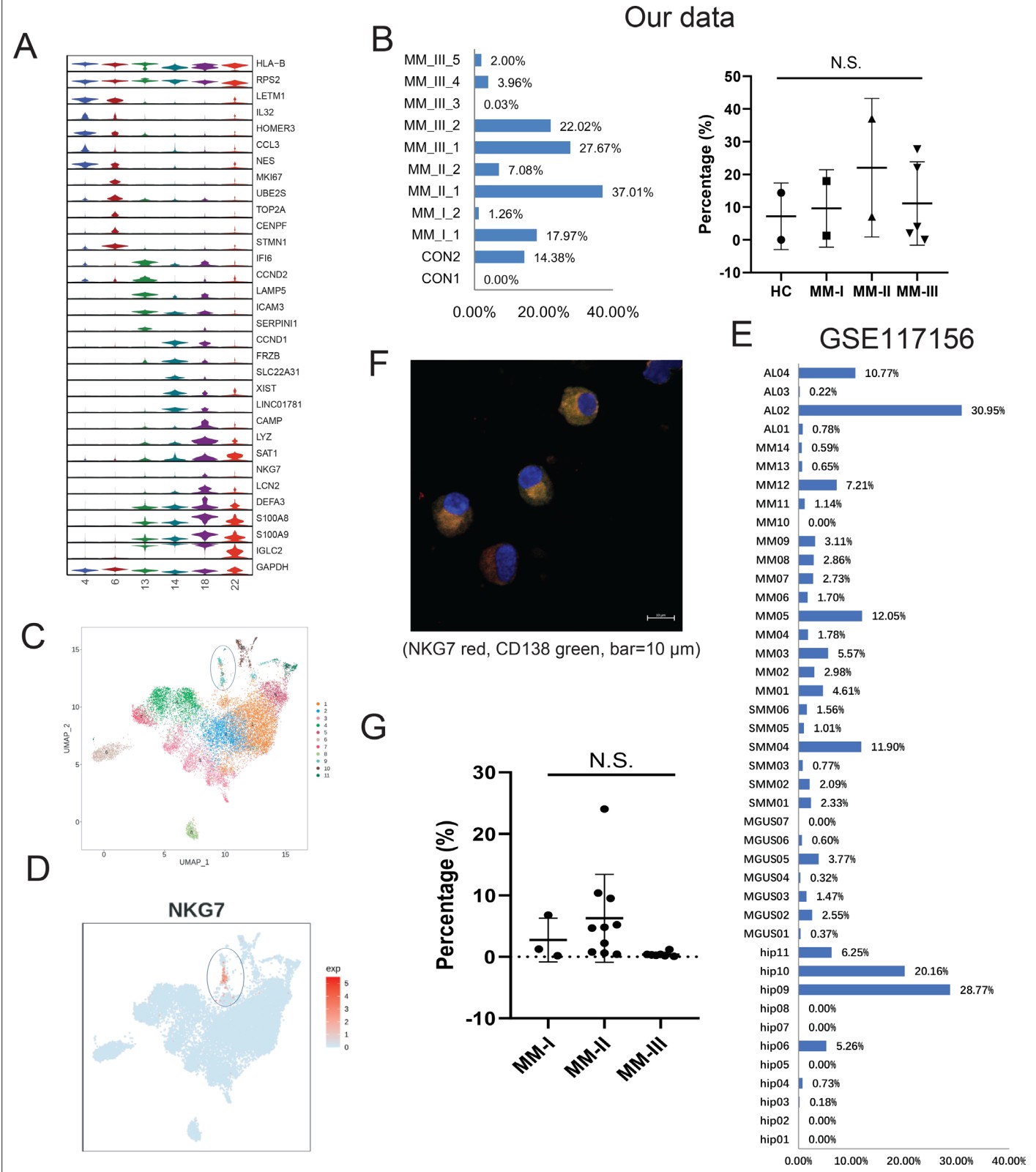

**Figure 2.** Identification of a rare cytotoxic NKG7⁺ plasma cell population in multiple myeloma (MM). (**A**) The heterogeneity of plasma cells (CD138+) was transcriptionally analysed, and genes specifically expressed in six plasma cell clusters were calculated. (**B**) In our cohort, the proportion of PC18 in all 11 plasma cells. (**C**) PC18 corresponds to c11 in another MM plasma dataset of GSE117156. (**D**) C11 specifically highly expresses cytotoxic markers like NKG7. (**E**) The cell fraction of c11 in another MM plasma dataset of GSE117156. (**F**) Immunofluorescence confirmed the expression of

*Figure 2 continued on next page*

*Figure 2 continued*

NKG7 in cytoplasm of myeloma cells (CD138 positive) from patients with MM; NKG7 red, CD138 green, bar = 10 μm. (**G**) Twenty MM patients (Revised International Staging System [R-ISS] stage I, three patients; II, 10 patients; III, seven patients) were enrolled in for multiparameter flow cytometry (MFC) analysis.

The online version of this article includes the following figure supplement(s) for figure 2:

**Figure supplement 1.** UMAP showing markers expression of CD138 and CD38 in multiple myeloma (MM).

**Figure supplement 2.** Multiparameter flow cytometry (MFC) analysis of NKG7+ plasma cells in MM-I, -II, and -III patients.

acquired and are shown in *Supplementary file 1*. Then, we performed functional network analysis with GluGo of these 351 DEGs. As shown in *Figure 5*, seven functional gene modules were identified: (a) ribosome, (b) protein processing in endoplasmic reticulum, (c) oxidative phosphorylation, (d) proteasome, and (e) EBV infection. It is not surprising that protein and energy metabolism-related modules, such as ribosomes in protein translation, protein processing in the endoplasmic reticulum, and proteasomes in protein degradation and oxidative phosphorylation, are enriched, which provides a synthetic basis for MM progression. Intriguingly, 18 genes involved in EBV infection attracted our attention. We then validated the expression of MKI67 and PCNA, two markers of proliferation, in MM patients. Indeed, significantly elevated expression of MKI67 and PCNA was observed in EBV-positive (EBV+) MM patients compared with EBV-negative (EBV-) MM patients (*Figure 5B, C*). Finally, potential therapeutic drugs were analysed based on pathways in PCC6, and alsterpaullone (cyclin-dependent kinase inhibitor) (*Watanabe et al., 2020*), orlistat (anti-obesity drug) (*Yang et al., 2010*), moxonidine (a selective imidazoline/alpha2 adrenergic receptor agonist) (*Molderings et al., 2003*), nalidixic acid (topoisomerase II inhibitors) (*Nyce, 1989*), and LY-294002 (PI3K/AKT inhibitor) (*Chen et al., 2018*) were proposed as C10-targeting pharmaceuticals in MM (*Figure 5D*).

## R-ISS stage-dependent expression analysis highlights functional modules underlying cytotoxic T cell decreases

Accumulating evidence demonstrated that the compromised tumour immune microenvironment (TIME) contributed to MM progression (*García-Ortiz et al., 2021*). And therapeutic agents targeting the TIME emerged as promising avenues for research (*Minnie and Hill, 2020*; *Kawano et al., 2015*). As one of the major cytotoxic immune cell types, T cell dysfunction was well known (*Cohen et al., 2020*). And immunotherapies such as chimaeric antigen receptor T cells (*Cohen et al., 2019*) and immune checkpoint inhibitors (*Chim et al., 2018*) have entered into clinical trials. Here, we proposed a hypothesis that the proportions of certain cytotoxic T cell populations decrease along with MM progression: we stratified them according to the R-ISS system and attempted to identify the functional genes.

First, re-clustering of T cells generated 21 clusters (*Figure 6A*), consisting of CD4+ T cells, CD8+ T cells, NK cells, and NKT cells (*Figure 6B and D*) and detailed expression of T cell markers was shown in *Figure 6—figure supplement 1*. No biased distribution was observed in 11 samples (*Figure 6C*). The percentages of T1 to T21 in MM versus healthy controls and MM R-ISS I–III were presented in *Figure 6E and F*, respectively. It was worthy that two clusters conformed with the hypothesis of decreased percentage along with R-ISS stages: T4 (cluster 4) and T6 (cluster 6). T4 was marked by high expression of CD8A and no expression of NKG7 and was identified as CD8+ T cells. T6 cells express both CD8A and NKG7 and was defined as NKT cells. We concentrate on T4 and T6 in subsequent work.

To identify R-ISS-dependent gene modules in T4 and T6, the R package MFUZZ (*Kumar and E Futschik, 2007*) was applied (*Figure 7—figure supplement 1*). In T4 CD8+ T cells, 12 gene modules with distinct expression patterns were generated, and module 7 (stable expression in healthy controls I-II and dramatic increases in stage III) and module 11 (gradually decreased expression with R-ISS stage) were chosen for subsequent analysis (*Figure 7A*). Genes in module 7 were functionally related to haemopoiesis, and genes in module 11 were involved in T-helper 1 cell lineage commitment (*Figure 7C*). For T6 NKT cells, genes in module 3 showed a similar expression trend to that of module 7 in T4 CD8+ T cells, and genes in module 7 showed the reverse trend (stable in healthy controls and R-ISS stages I-II and dramatic increases in stage III) (*Figure 7B*). Subsequent ClueGo results (*Figure 7D*) showed that genes involved in regulation of megakaryocyte differentiation and regulation of myeloid

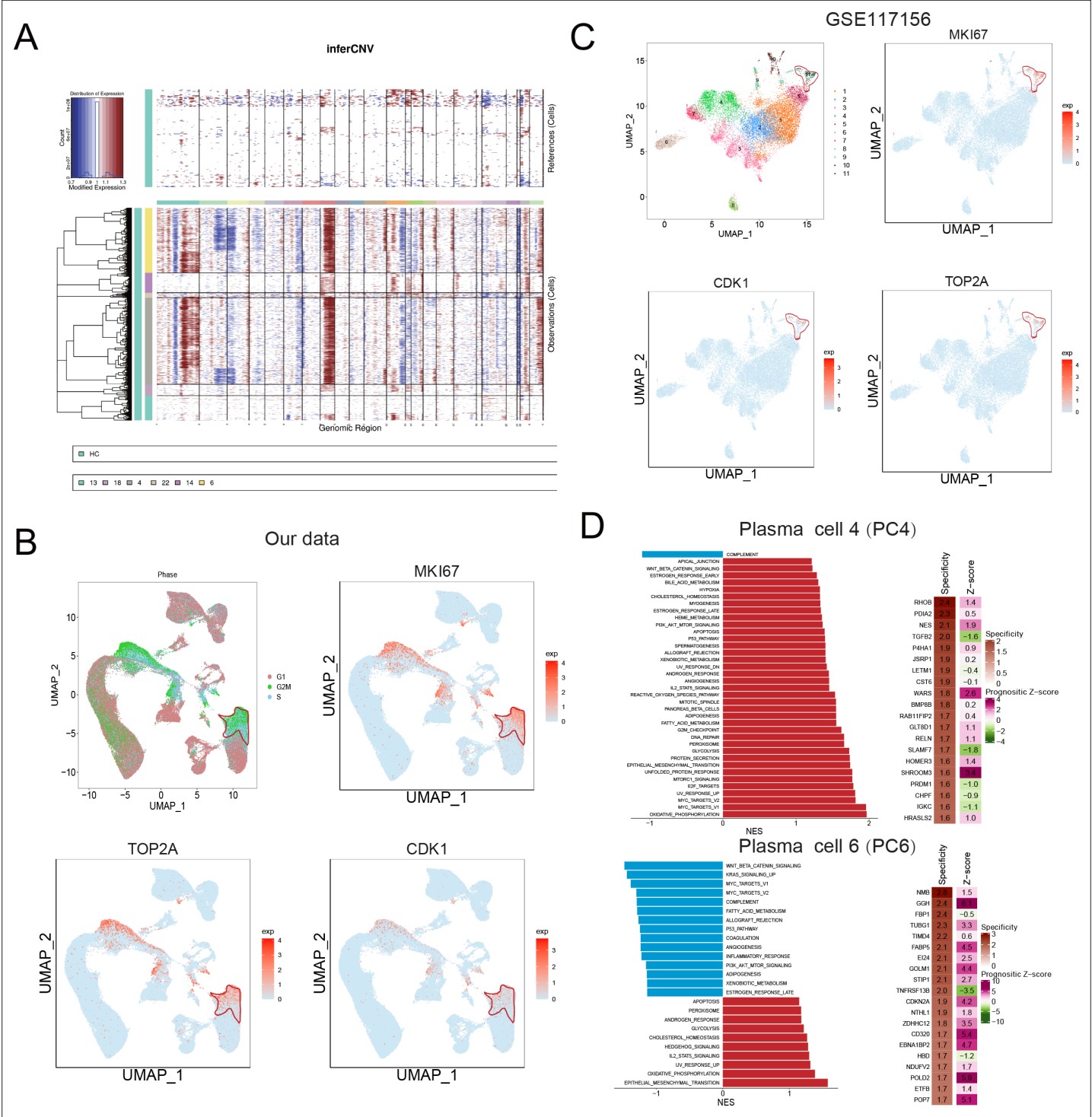

**Figure 3.** Identification of a malignant and risky plasma cell cluster with proliferation activity. (**A**) To identify malignant plasma cells, inferCNV was applied to obtain the copy number variation (CNV) signals in all six plasma cell clusters. (**B**) Cell cycle distribution was analysed, and three proliferation markers of MKI67, TOP2A, and CDK1 were also shown. (**C**) UMAP of three proliferation markers of MKI67, TOP2A, and CDK1 in GSE117156. (**D**) GSEA hallmarks and PRECOG analysis was conducted, and significant enriched pathways of clusters 4 and 6 were shown. PRECOG was applied to infer the genes in clusters 4 and 6 with specificity and prognostic z-score in external multiple myeloma (MM) gene expression data GSE6647.

The online version of this article includes the following figure supplement(s) for figure 3:

**Figure supplement 1.** The inferCNV result of B and plasma cell clusters in MM-III stage samples.

**Figure supplement 2.** GSEA and PRECOG analysis of plasma clusters 13, 14, 18, and 22.

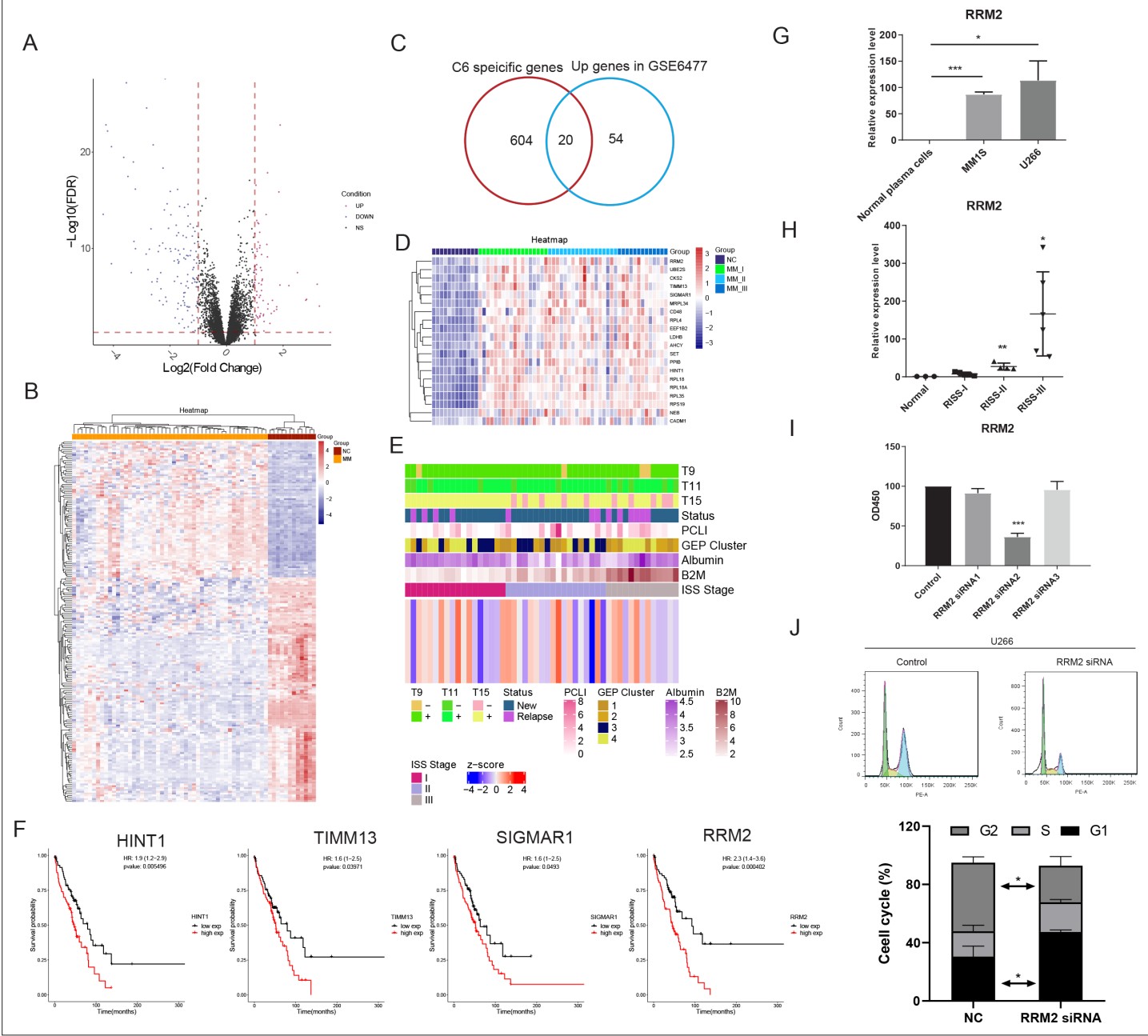

**Figure 4.** Bulk sequencing validation highlights PCC6-specific genes ribonucleotide reductase regulatory subunit M2 (RRM2) as novel prognostic markers in multiple myeloma (MM). All 75 significant deregulated genes in GSE6647 dataset MM group was acquired, and shown with (**A**) volcano plot and (**B**) heat map. (**C**) Specific genes with markers potential was compared with 75 differentially expressed genes (DEGs) in MM dataset of GSE6647, and 20 genes were acquired and named as deregulated proliferating marker genes in MM (DPMGs). (**D**) The expression of 20 DPMGs in normal and R-ISS I–III stages were shown, and all seven DPMGs were up-regulated in MM, especially in R-ISS III stage. (**E**) Clinical parameters of MM samples with 20-gene signature score. The T9/11/15 in **E** referred to the trisomies of chromosomes 9, 11, and 15. (**F**) Four DPMGs, SIGMAR1, TIMM13, RRM2, and HINT1, exhibited performance as unfavourable prognostic markers in MM patients. (**G**) Relative quantification of RRM2 by qRT-PCR in normal plasma cells and MM1S and U266 cell lines. (**H**) Relative quantification of RRM2 by qRT-PCR in healthy and R-ISS stratified MM patient BM samples. (**I**) Proliferation phenotype of RRM2 silencing in MM cell line U266. (**J**) RRM2 silencing caused MM cells arrest in G1 phase. Values represent the means of three experiments ± SD; *p<0.05, **p<0.01, versus untreated control.

The online version of this article includes the following figure supplement(s) for figure 4:

**Figure supplement 1.** Expression and in vitro cellular functions of histidine triad nucleotide binding protein 1 (HINT1) silencing in myeloma U266 cell lines.

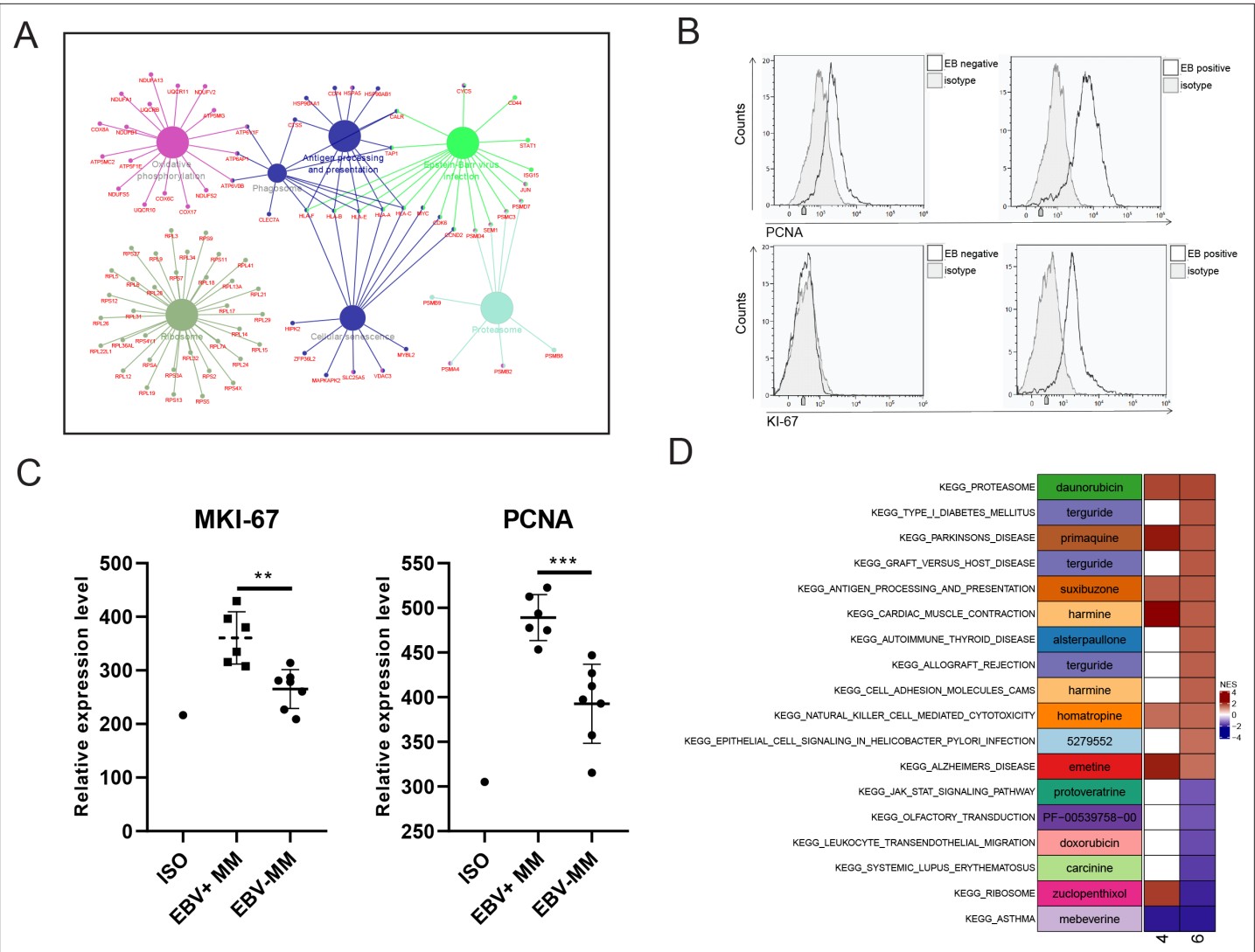

**Figure 5.** Proliferating plasma cells were increased in Epstein-Barr virus (EBV)-positive multiple myeloma (MM) patients. (**A**) Differentially expressed genes (DEGs) in MM R-ISS III stage with I stage were obtained and conferred to functional analysis. (**B**) Relative expression of MKI-67 and PCNA in EBV-positive and EBV-negative MM patients BM samples. (**C**) Representative FACS peaks of MKI-67 and PCNA in EBV-positive and EBV-negative MM patients. (**D**) Based on pathways enriched in plasma clusters 4 and 6, potential drug candidates were acquired, and shown with heat map.

cell differentiation were enriched in module 3, and module 7 genes were involved in leukocyte activation involved in the immune response.

## Ligand-receptor pairs and potential immunotherapeutic targets in CD8+ T-neutrophil and CD8+ T/NKT-PC communication

Finally, we employed CellPhoneDB (*Efremova et al., 2020*) to interrogate cell-cell communication between T4 CD8+ T cells and T6 NKT cells with myeloid cell clusters and proliferating plasma PC6 cells, respectively. As shown in *Figure 8A*, the myeloid C9 cluster showed the most ligand-receptor pairs with T4 CD8+ T cells (69) and T6 NKT cells (71). The ligand-receptor pairs between T4 CD8+ T cells and myeloid C9 cluster included IL15-IL15 receptors and phagocytic checkpoints SIRPA-CD47. In addition to SIRPA-CD47, well-known ligand-receptor pairs between T6 NKT cells and myeloid C9 cluster included CCL5-CCR1 and LGALS9-CD44 (*Figure 8C* and *Figure 8—figure supplement 1*). For T2 CD8+ T cell/T10 NKT cell communication with PC6 proliferating PCs (*Figure 8B*), common ligand-receptor pairs such as CD74-MIF, ADRB2-VEGFB, and CCL5-CCR1 were identified (*Figure 8D* and *Figure 8—figure supplement 2*). Altogether, the ligand-receptor pairs between T cells communicating

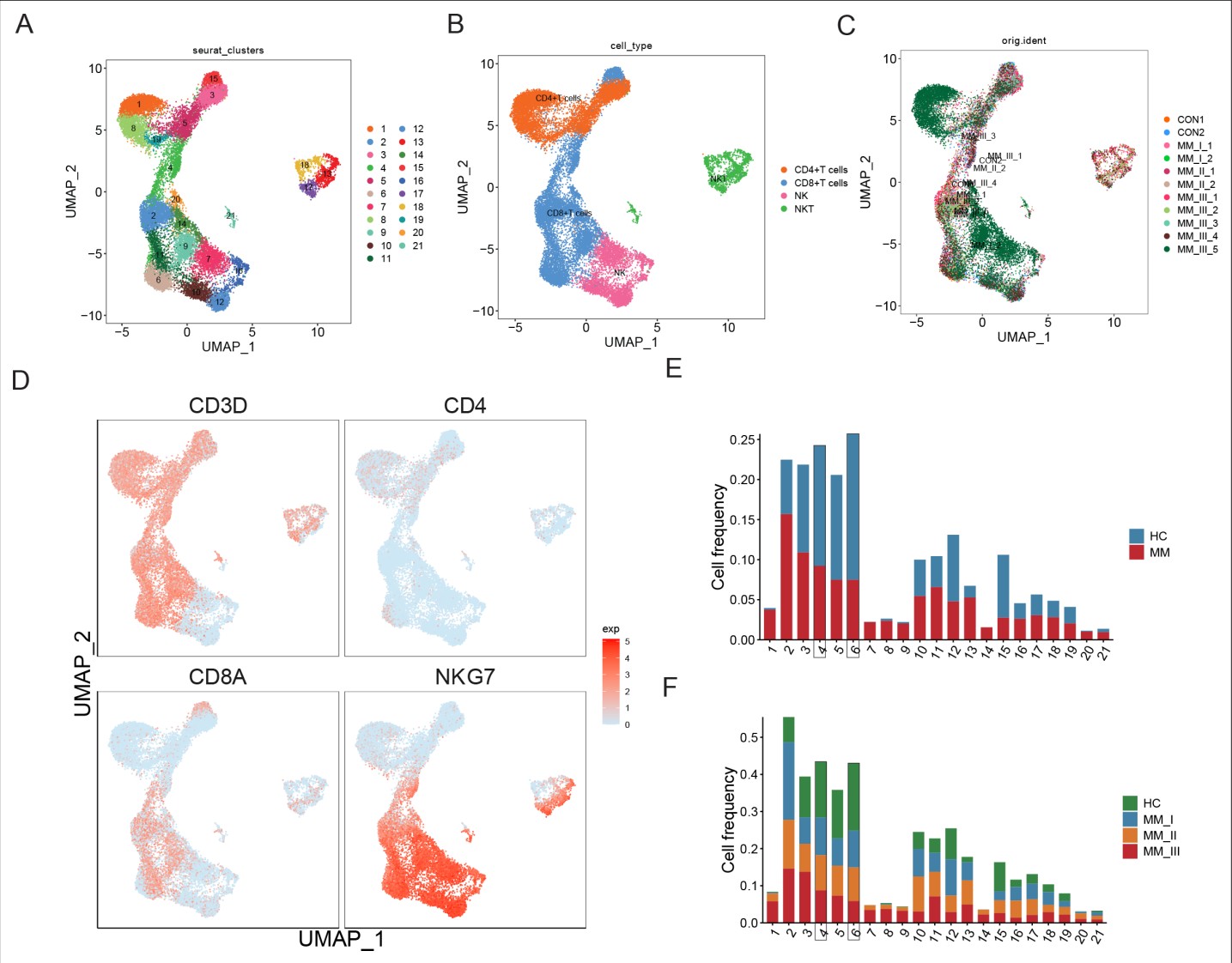

**Figure 6.** T cell population analysis suggested stage-dependent CD8+ T and NKT cell clusters depletion in multiple myeloma (MM). (**A**) T cells and NK cells were reclustered, and 21 clusters were acquired. (**B**) Based on markers expression, CD4+ T, CD8+ T, NK, and NKT cells were shown with UMAP. (**C**) UMAP of samples distribution of T cells and NK cells. (**D**) Markers of T cells and NK cells, shown with UMAP. (**E**) Cluster proportions in healthy control and MM groups. (**F**) Cluster proportions in healthy control, R-ISS I, II, and III MM groups.

The online version of this article includes the following figure supplement(s) for figure 6:

**Figure supplement 1.** Violin plot showing expression of T cell markers.

with myeloid and proliferating PCs suggested a complex communication network in the MM TIME, which could provide clues for MM progression and therapy.

## Discussion

MM is characterized by uncontrolled proliferation of monoclonal PCs, and the R-ISS system was developed to stratify MM patients into stage I, II, and III (*Palumbo et al., 2015*) with distinct outcomes (*Kastritis et al., 2017*) and treatment response (*Cho et al., 2017*). Several studies on scRNA-seq in MM have been disclosed some heterogeneity of MM, which hint us to explore further. The R-ISS staging system is an internationally wide used prognostic stratification system. However, the difference among different stages inducing clearly different outcomes has still been unclear, and urgent to be solved for precision therapy. Therefore, we focused on the differences among the three stages. This

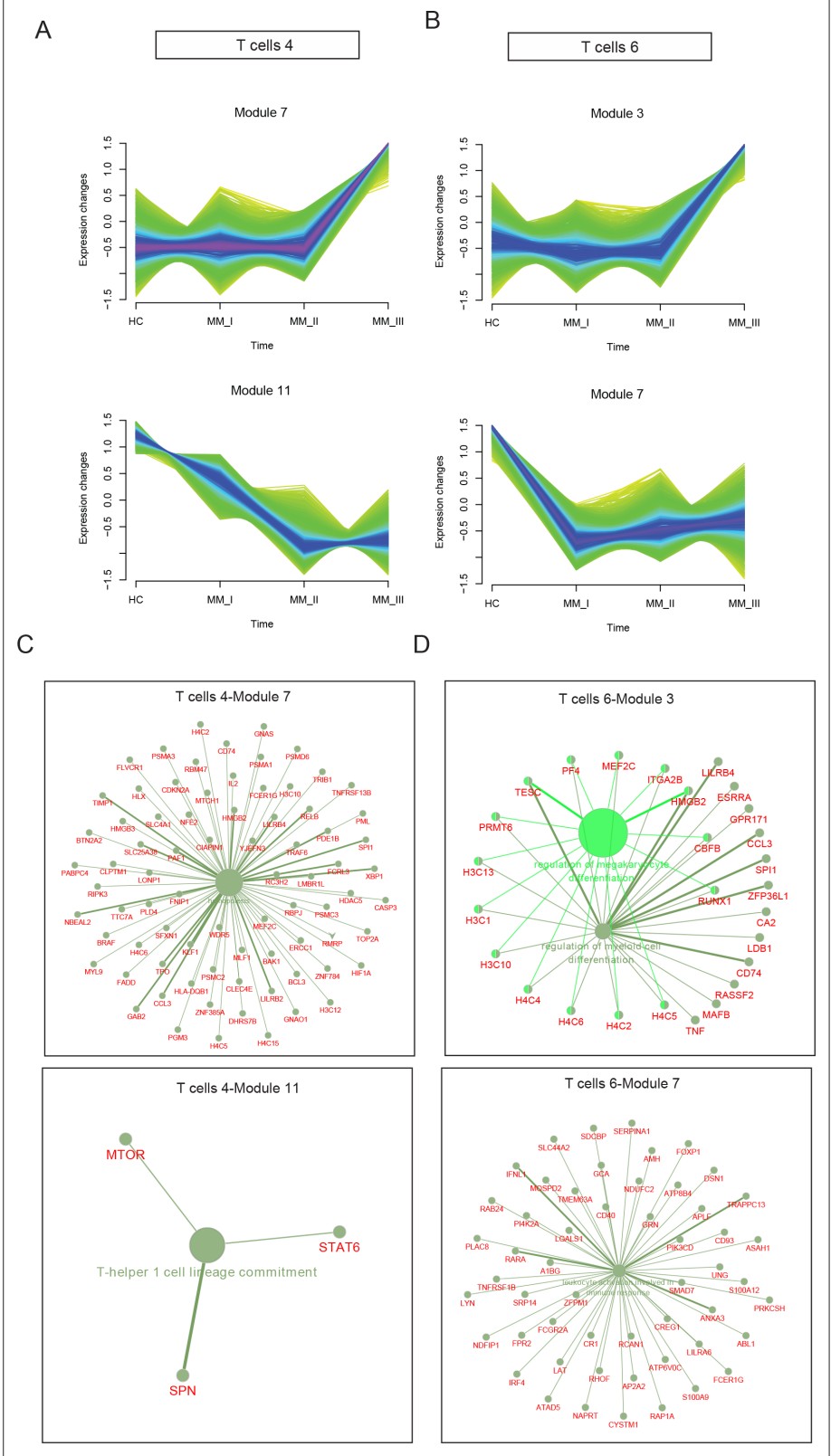

**Figure 7.** Stage-dependent expression analysis reveals gene modules in CD8+ T and NKT cell clusters. To acquire Revised International Staging System (R-ISS) stage-dependent gene expression modules in cluster 4 and 6 populations, MFUZZ was acquired. (**A**) (Upper) Genes in Cluster 4T cells-Module 7 (**T4–M7**) were generally characterized as stable expression in healthy controls I-II and dramatic increases in III stage. (Lower) Genes in

*Figure 7 continued on next page*

*Figure 7 continued*

Cluster 4T cells-Module 11 (**T4–M11**) were generally marked with gradually decreased expression with R-ISS stage. (**B**) (Upper) Genes in Cluster 6T cells-Module 3 (**T6–M3**) were generally characterized as stable expression in healthy controls I-II and dramatic increases in III stage. (Lower) Genes in Cluster 6T cells-Module 7 (**T6–M7**) were generally characterized as stable in healthy controls, R-ISS I-II and dramatic increases in III stage. (**C**) (Upper) Functional network of genes in T4-M7 highlights haemopoiesis and (lower) functional network of genes in T4-M11 suggested T-helper 1 cell lineage commitment. (**D**) (Upper) Functional network of genes in T6-M3 indicates genes in megakaryocyte differentiation and regulation of myeloid cell differentiation. (Lower) Functional network of genes in T6-M7 indicates genes in leukocyte activation.

The online version of this article includes the following figure supplement(s) for figure 7:

**Figure supplement 1.** MFUZZ analysis identifies 12 modules in each of T4 and T6 T cells.

study identified malignant PCs with potent proliferation ability. Moreover, RRM2, the marker of proliferating PCs with unfavourable prognostic significance in MM, was also characterized. RRM2, regulated in a cyclin F-dependent fashion (***D'Angiolella et al., 2012***), encodes one of two non-identical subunits for ribonucleotide reductase and is well studied as an oncogene and poor prognostic marker in multiple solid cancer types (***Mazzu et al., 2019***; ***Gandhi et al., 2020***; ***Rahman et al., 2013***). Meanwhile, RRM2 inhibition acted in synergy with gemcitabine in lymphoma (***Jones et al., 2011***) and with a WEE1 inhibitor in H3K36me3-deficient cancers (***Pfister et al., 2015***). In MM, RRM2 knockdown alone inhibits MM cell proliferation and induces apoptosis via the Wnt/β-catenin signalling pathway (***Liu et al., 2019***). In contrast, HINT1 encodes a protein that hydrolyses purine nucleotide phosphoramidate substrates and is believed to be act mainly as a tumour suppressor in multiple cancer types (***Jung et al., 2020***; ***Motzik et al., 2017***; ***Wang et al., 2007***). The tumour-suppressing effect of RRM2

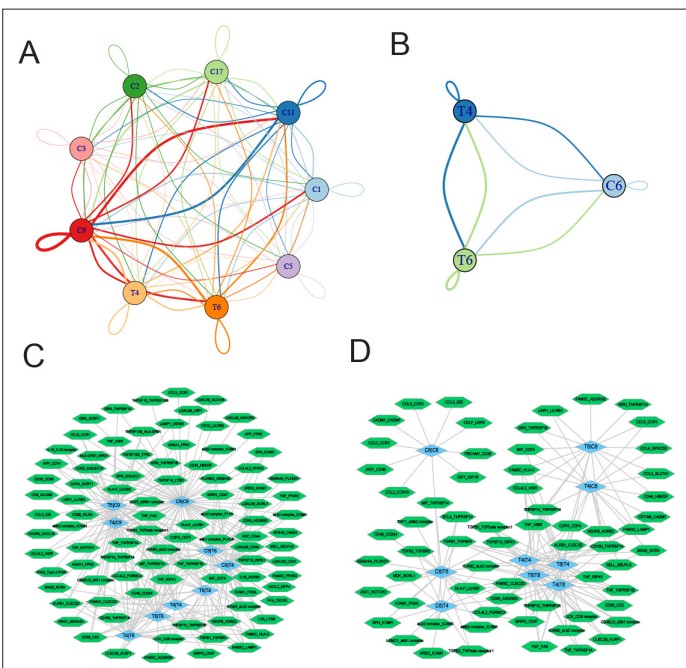

**Figure 8.** Ligand-receptor pairs and potential immunotherapeutic targets in CD8+ T-myeloid and CD8+ T/NKT-plasma cells communication. (**A**) Cell-cell communications showing the interaction numbers between myeloid cells and T4 CD8+ T and T6 NKT cells. (**B**) Paracrine ligand-receptor interaction pairs between myeloid cells and T4 CD8+ T and T6 NKT cells. (**C**) Cell-cell communications showing the interaction numbers between PC6 plasma cells and T4 CD8+ T, T6 NKT cells. (**D**) Paracrine ligand-receptor interaction pairs between PC6 plasma cells and T4 CD8+ T, T6 NKT cells.

The online version of this article includes the following figure supplement(s) for figure 8:

**Figure supplement 1.** Ligand-receptors interaction pairs between T cells with myeloid cells.

**Figure supplement 2.** Ligand-receptor interaction pairs between T cells with proliferating plasma cells.

silencing may provide a promising translational target for MM. Further in-depth studies of their roles in PC proliferation and MM are needed in future.

Intriguingly, we also found a rare PCs proportion with cytotoxic activities (high expression of NKG7) in the BM microenvironment of MM patients for the first time. T cells and NK cells are the two main types of cytotoxic immune cells in previous studies. Coincidentally, plasma/B cells producing GZMB also possess cytotoxic activities and induce HCT-116 cell death (*Cupi et al., 2014*). To be cautious, we first validated the existence of these cytotoxic NKG7+ PCs in another single-cell dataset. In the future, in vitro and in vivo cytotoxicity assays should be conducted to study the function of NKG7+ PCs in inducing cell death in MM. Altogether, this discovery offers an alternative option for the cytotherapy of MM.

In addition, we also identified multiple immunotherapeutic targets in MM. CD24 is a highly expressed, anti-phagocytic signal in several cancers and demonstrates therapeutic potential for CD24 blockade in cancer immunotherapy (*Barkal et al., 2019*). ICAM-1 antibodies exhibit potent anti-myeloma activity in multiple studies (*Veitonmäki et al., 2013*; *Sherbenou et al., 2020*; *Gu et al., 2019*). CD44 mediates resistance to lenalidomide in MM (*Bjorklund et al., 2014*), and CD44-targeted T cells mediate potent anti-tumour effects against MM (*Casucci et al., 2013*). CD47 on macrophages represents a 'do-not-eat-me' immune checkpoint (*Liu, 2019*), and targeting CD47 has been proposed as a novel immunotherapy for MM. Macrophage migration inhibitory factor (MIF) is an important player and a novel therapeutic target in MM, and inhibiting MIF activity will sensitize MM cells to chemotherapy (*Zheng et al., 2016*). MIF plays a crucial role in MM sensitivity to protease inhibitors and suggests that targeting MIF may be a promising strategy to (re)sensitize MM to treatment (*Sun et al., 2020*). Together, these identified ligand-receptor interactions can provide options for MM immunotherapy.

There were some limitations to this study. First, the functions of NKG7+ and proliferating PCs need to be further studied. Although the two plasma subtypes showed no significant difference in R-ISS stage, whether they exerted tumour-targeting cytotoxic and initiating functions in MM requires further in vitro and in vivo evidence. Next, the clinical expression, prognosis, and loss-of-function cellular phenotypes of the two oncogenes, RRM2 and HINT1, were interrogated, and more data on the in vivo and underlying mechanism will provide a more comprehensive perspective. Finally, through ligand-receptor interaction analysis, we identified a CD47-SIRPA-mediated phagocytic checkpoint between plasma and T cells, which suggests an immune surveillance defect with MM progression, and corresponds to clinical studies.

In conclusion, we constructed a single-cell transcriptome atlas of BM in normal health controls and R-ISS-staged MM patients. Focusing on PCs, we identified and validated the existence of NKG7+ cytotoxic PCs. In addition, a malignant PC population with high proliferation capability (proliferating PCs) was clinically associated with an unfavourable prognosis. RRM2, a specific marker of proliferating PCs, was shown to induce MM cell line proliferation and serve as a detrimental marker in MM. Subsequently, three R-ISS-dependent gene modules in cytotoxic CD8+ T and NKT cells were identified and functionally analysed. Finally, cell-cell communication between myeloid cells and proliferating PCs with cytotoxic CD8+ T and NKT cells was investigated, which identified intercellular ligand receptors and potential immunotargets such as SIRPA-CD47. Collectively, the results of this study provided an R-ISS-related single-cell MM atlas and revealed the clinical significance of two PC clusters, as well as potential immunotargets in MM progression.

## Materials and methods
### Patients and sample collection

This study included nine MM patients diagnosed as active MM according to International Myeloma Working Group guideline, and two age-matched normal control (transplant donor).

For FACS analysis of MKI67, PCNA and NKG7 in PCs, samples from 3 healthy donors, 3 R-ISS I, 10 R-ISS-II, and 7 R-ISS III were collected. Written informed consents were obtained from all subjects. All experimental procedures were approved by the Medical Ethics Committee of Sichuan Provincial People's Hospital and carried out in accordance with the principles of the Declaration of Helsinki. ALL of the patients signed the informed consent containing contact information, the purpose of the study, risks and benefits from this study, consent to publish the manuscript.

BM aspirates were collected into EDTA-containing tubes, and lysed using Versalyse Lysing Solution (cat. no. A09777; Beckman Coulter, Inc, Brea, CA, USA). Mononuclear cells were isolated using a Ficoll gradient (density 1.077 g/ml, cat. no. 07801; STEMCELL Technologies, Canada). Fresh single-cell suspensions were used for scRNA-seq. Aliquots of the same bone biopsy were analysed by fluorescence in situ hybridization and MFC (Navios, Beckman Coulter, Inc, Brea, CA, USA) as parts of the routine clinical diagnosis. In MFC, cell populations were considered abnormal if they have an atypical differentiation pattern, an increased or decreased expression level of normal antigens, an asynchronous maturational pattern or express aberrant antigens (*Flores-Montero et al., 2017*).

## scRNA-seq library construction and sequencing

scRNA-seq libraries were prepared with Chromium Single cell 3' Reagent v3 Kits according to the manufacturer's protocol. Single-cell suspensions were loaded on the Chromium Single Cell Controller Instrument (10× Genomics) to generate single-cell gel beads in emulsions (GEMs). Briefly, about $2×10^5$ PBMC single cells were suspended in calcium- and magnesium-free PBS containing 0.04% weight/volume BSA. About 22,000 cells were added to each channel with a targeted cell recovery estimate of 10,000 cells. After generation of GEMs, reverse transcription reactions were engaged barcoded full-length cDNA followed by the disruption of emulsions using the recovery agent and cDNA clean up with DynaBeads MyOne Silane Beads (Thermo Fisher Scientific, New York, NY, USA). cDNA was then amplified by PCR with appropriate cycles which depend on the recovery cells. Subsequently, the amplified cDNA was fragmented, end-repaired, A-tailed, index adaptor ligated, and library amplification. Then these libraries were sequenced on the Illumina sequencing platform (NovaSeq6000) and 150 bp paired-end reads were generated. The GEM generation, library construction, and sequencing were performed by OE Biotech Co., Ltd (Shanghai, China).

## scRNA-seq data processing

The Cell Ranger software pipeline (version 3.1.0) provided by 10× Genomics was used to demultiplex cellular barcodes, map reads to the genome and transcriptome using the STAR aligner, and downsample reads as required to generate normalized aggregate data across samples, producing a matrix of gene counts versus cells. We processed the unique molecular identifier (UMI) count matrix using the R package Seurat (*Haghverdi et al., 2018*) (version 3.1.1). To remove low-quality cells and likely multiplet captures, which is a major concern in microdroplet-based experiments, we applied criteria to filter out cells with UMI/gene numbers out of the limit of mean value ±2-fold of standard deviations assuming a Gaussian distribution of each cells' UMI/gene numbers (*Liu et al., 2021*; *Lu et al., 2020*). In scale function, the regression out method we used was ScaleData object = seurat_ob, vars. to.regress=c('nCount_RNA', 'percent.mito'). Following visual inspection of the distribution of cells by the fraction of mitochondrial genes expressed, we further discarded low-quality cells where >20% of the counts belonged to mitochondrial genes. After applying these QC criteria, 103,043 single cells were retained for downstream analyses. Library size normalization was performed with Normalize Data function in Seurat (*Haghverdi et al., 2018*) to obtain the normalized count. Specifically, the global-scaling normalization method 'LogNormalize' normalized the gene expression measurements for each cell by the total expression, multiplied by a scaling factor (10,000 by default), and the results were logtransformed.

Top variable genes across single cells were identified using the method described in *Macosko et al., 2015*. The most variable genes were selected using FindVariableGenes function(mean.function=ExpMean, dispersion.function=LogVMR) in Seurat (*Haghverdi et al., 2018*). To remove the batch effects in scRNA-seq data, the mutual nearest neighbors presented by Haghverdi et al. was performed with the R package batchelor (*Haghverdi et al., 2018*). Graph-based clustering was performed to cluster cells according to their gene expression profile using the FindClusters function in Seurat (*Haghverdi et al., 2018*). We applied Scrublet (*Wolock et al., 2019*) to computationally infer and remove doublets in each sample individually, with an expected doublet rate of 0.06 and default parameters used otherwise. The doublet score threshold was set by visual inspection of the histogram in combination with automatic detection. Cells were visualized using a two-dimensional t-distributed stochastic neighbor embedding algorithm with the RunTSNE function in Seurat (*Haghverdi et al., 2018*). We used the FindAllMarkers function(test.use=bimod) in Seurat (*Haghverdi et al., 2018*) to identify marker genes of each cluster. For a given cluster, FindAllMarkers identified positive markers

compared with all other cells. Then, we used the R package SingleR (*Aran et al., 2019*), a novel computational method for unbiased cell type recognition of scRNA-seq, with the reference transcriptomic datasets 'Human Primary Cell Atlas' (*Mabbott et al., 2013*) to infer the cell of origin of each of the single cells independently and identify cell types. The clusters of T cells were annotated with naïve T cell markers, inhibitory cell markers, cytokine and effector molecules, co-stimulatory molecules, transcription factors, and Tregs markers, according to Guo et al's study (*Guo et al., 2018*).

DEGs were identified using the FindMarkers function (test.use=MAST) in Seurat (*Haghverdi et al., 2018*). p-Value <0.05 and |log₂foldchange|>0.58 was set as the threshold for significantly differential expression. Gene Ontology (GO) enrichment and KEGG pathway enrichment analysis of DEGs were respectively performed using R based on the hypergeometric distribution.

## Large-scale chromosomal CNV analysis

The normalized scRNA-seq gene expression matrices were used to estimate CNV profiles with inferCNV R package (*Vogelstein et al., 2013*). Genes were sorted based on their chromosomal location and a moving average of gene expression was calculated using a window size of 101 genes. The expression was then centred to zero by subtracting the mean. The de-noising was carried out to generate the final CNV profiles.

## Cell culture

Human normal PCs were separated from the peripheral blood of healthy donors using flow cytometry. All donors were healthy volunteers who had not previously received any drugs associated with immunological diseases. Briefly, peripheral blood mononuclear cells were separated by Ficoll-Hypaque centrifugation from peripheral blood. PCs were isolated from mononuclear blood cells using CD138 microbeads (Miltenyi Biotec, Inc, Germany) according to the manufacturer's instructions (*Horst et al., 2002*). The isolated PCs were then cultured. The U266 and MM1.S cell lines were purchased from the American Type Culture Collection. RPMI-1640 medium (Gibco; Thermo Fisher Scientific, Inc, Waltham, MA, USA) containing 10% fetal bovine serum (Gibco; Thermo Fisher Scientific, Inc, Waltham, MA, USA) and 1% dual antibiotics (penicillin 100 U/ml; streptomycin 0.1 mg/ml; Sigma-Aldrich; Merck KGaA) was used to culture all cells at 37°C with 5% $CO_2$.

## RRM2 and HINT1 silencing with shRNAs

Short hairpin RNAs (shRNAs) specifically targeting RRM2 and HINT1 were designed and synthesized by RiboBio(Guangzhou, China). RRM2 shRNAs were shRNA1 (5'–3'): GGAGCGATTTAGCCAAGAA, shRNA2 (5'–3'): GCCTCACATTTTCTAATGA and shRNA3 (5'–3'): GAAAGACTAACTTCTTTGA. HINT1 shRNAs were: shRNA1 (5'–3'): GGTGGTGAATGAAGGTTCA, shRNA2 (5'–3'): GTGATACCCAAG AAACATA and shRNA3 (5'–3'): GTCTGTCTATCACGTTCAT.

The shRNAs were cloned into pcDNA3.1, and transfected into the MM cells. Transfection was performed according to the manufacturer's protocol using Lipofectamine 3000 reagent. Following co-culture for 12 hr at 37°C, the medium was replaced with culture media and the transfected cells were used for subsequent experiments. Forty-eight hours post transfection, RRM2 and HINT1 expression in the cells was determined by reverse transcription quantitative polymerase chain reaction (RT-qPCR), and the transfection efficiency was verified.

## RNA extraction and RT-qPCR

Total RNA from cells was extracted using TRIzol (Invitrogen; Thermo Fisher Scientific, Inc, Waltham, MA, USA). cDNA was synthesized using the PrimeScript RT reagent kit (Takara Biotechnology Co., Ltd., Dalian, China) at 25°C for 5 min, 37°C for 30 min and 85°C for 5 s. qPCR analysis was performed using the StepOnePlus Real-Time PCR system (Applied Biosystems; Thermo Fisher Scientific, Inc, Carlsbad, CA, USA) with SYBR Premix EX Taq kit (Takara Biotechnology Co., Ltd., Dalian, China). Primer sequences for RRM2, HINT1, were as follows: RRM2 Forward 5'–3': CCAATGAGCTTCACAG GCAA, Reverse 5'–3': TGGCTCAAGAAACGAGGACT. HINT1Forward5'–3':TTGCCGACCTCCAAGA ACAT, Reverse 5'–3': CCCTCAAGCACCAACACATT. Relative expression was calculated using the 2−ΔΔCq method (*Livak and Schmittgen, 2001*).

## CCK8 assays for cell proliferation

Cell proliferation assay was performed with Cell Counting Kit-8 (Dojindo, Kumamoto, Japan) according to the manufacturer's instruction. Twenty-four hour after transfection, cells were seeded in 96-well

plates at 1×10⁴ U266 cells per well. The proliferative ability of U266 cells was determined at 0, 24, and 48 hr. The absorbance was measured at 450 nm using a microplate spectrophotometer (Molecular Devices, Sunnyvale, CA, USA).

## Cell cycle analysis

Cell cycle genes were defined as those with a 'cell cycle process' GO annotation (downloaded from MSigDB version 3.1). We defined four cell cycle signatures (G1/S, S, G2/M, and M) as the average expression [log2(TPM+1)] of phase-specific subsets of the cell cycle genes. We refined these signatures by averaging only over those genes whose expression pattern in our data correlated highly ($r > 0.5$) with the average signature of the respective cell cycle phase (before excluding any gene) in order to remove the influence of genes.

## Immunofluorescence staining

To determine whether NKG7 expressed in myeloma cells, immunofluorescence staining was performed. The BM smears were fixed with 4% paraformaldehyde for 10 min, and the cells were incubated in a blocking solution containing 10% donkey serum with 0.25% Triton X-100 for 1 hr at room temperature. Primary antibodies (CD138 Santa Cruz sc-390791, and NKG7 Cell Signaling Technology E6S2A) were incubated overnight at 4°C followed by secondary antibodies for 1 hr at room temperature. DAPI was used for nuclear counterstaining (Molecular Probes, Eugene, OR, USA), and fluorescence images were acquired with an LSM 900 confocal microscope (ZEISS).

## Multiparameter flow cytometry

BM aspirates were collected with EDTA-containing tubes and analysed by MFC (Navios, Beckman Coulter, Brea, CA, USA) to distinguish the differences in the percentage of NKG7-positive cells among different R-ISS stages. In MFC, PCs were identified for CD38 and CD138 positive. And cell populations were considered abnormal if they displayed monoclonal cyto light chain. The isotypes were used to identify negative gates. Based on isotypes, NKG7-positive populations were identified. MFC for MKI67 (abcam, ab16667) and PCNA (abcam, ab18197) were similar to NKG7.

## Statistical analysis

Statistical analysis and graph representations were performed using SPSS version 13.0 software (SPSS Inc, Chicago, IL, USA) and GraphPad Prism 8 Software (GraphPad, San Diego, CA, USA), respectively. One-way ANOVA with post hoc Tukey's test was used to compare differences between multiple groups. For cell assays, data are presented as the mean ± standard deviation (SD) and were compared using either Student's t test or the Mann-Whitney U test. The Kaplan-Meier method was used for survival analyses. $p < 0.05$ was considered statistically significant.

## Acknowledgements

We thank all the patients and their families for participating in this study. The present study was supported by the grant from the National Natural Science Foundation of China (82002212), the Science & Technology Department of Sichuan Province (2022JDTD0024, 2022YFS0100, 2022YFS0334, 2021JDGD0043), the Chengdu Science and Technology Bureau (2019-YF05-00572-SN and 2022-YF05-01625-SN), the Sichuan cadre health care project (2022-216), the China Postdoctoral Science Foundation Grant (2019M663567), the foundation of Basic scientific research in Central UESTC (ZYGX2020J024), and Medicine-engineering interdisciplinary grant by UESTC (ZYGX2021YGLH006, ZYGX2021YGLH204).

# Additional information

## Funding

| Funder | Grant reference number | Author |
|---|---|---|
| National Natural Science Foundation of China | 82002212 | Ling Zhong |
| Chengdu Science and Technology Bureau | 2019-YF05-00572-SN | Ling Zhong |
| China Postdoctoral Science Foundation | 2019M663567 | Ling Zhong |
| University of Electronic Science and Technology of China | Basic scientific research ZYGX2020J024 | Ling Zhong |
| University of Electronic Science and Technology of China | Medicine-engineering interdisciplinary ZYGX2021YGLH006 | Ling Zhong |
| Department of Science and Technology of Sichuan Province | 2022JDTD0024 | Bo Gong |
| Chengdu Science and Technology Bureau | Applied Basic Research 2022-YF05-01625-SN | Bo Gong |
| Sichuan cadre health care project | Sichuan cadre health care project 2022-216 | Wei Zhang |
| University of Electronic Science and Technology of China | Medicine-engineering interdisciplinary ZYGX2021YGLH204 | Wei Zhang |
| Applied Basic Research | 2022YFS0100 | Peng Hao |
| Department of Science and Technology of Sichuan Province | 2022YFS0334 | Wei Zhang |
| Department of Science and Technology of Sichuan Province | 2021JDGD0043 | Wei Zhang |
| Department of Science and Technology of Sichuan Province | 2022YFS0100 | Jiang Hu |

The funders had no role in study design, data collection and interpretation, or the decision to submit the work for publication.

## Author contributions

Ling Zhong, Wei Zhang, Bo Gong, Conceptualization, Resources, Data curation, Software, Formal analysis, Supervision, Funding acquisition, Validation, Investigation, Visualization, Methodology, Writing – original draft, Project administration, Writing – review and editing; Peng Hao, Resources, Formal analysis, Methodology, Writing – original draft; Qian Zhang, Resources, Formal analysis, Validation, Investigation, Methodology; Tao Jiang, Resources, Validation, Investigation, Writing – original draft; Huan Li, Validation, Visualization; Jialing Xiao, Validation, Visualization, Methodology; Chenglong Li, Lan Luo, Validation; Chunbao Xie, flow cytometry; Jiang Hu, Funding acquisition, Investigation, Resources; Liang Wang, Supervision, Validation, Methodology; Yuping Liu, Conceptualization, Resources, Supervision, Validation, Methodology; Yi Shi, Conceptualization, Resources, Formal analysis, Supervision, Investigation, Methodology

## Author ORCIDs

Huan Li http://orcid.org/0000-0001-6624-9755
Bo Gong http://orcid.org/0000-0003-2763-6829

## Ethics

Human subjects: Written informed consents were obtained from all subjects. All experimental procedures were approved by the Medical ethics committee of Sichuan Provincial People's Hospital and carried out in accordance with the principles of the Declaration of Helsinki. ALL of the patients signed the informed consent containing contact Information, the purpose of the study, risks and benefits from this study, consent to publish the manuscript. The protocol numbers was 2020-240.

### Decision letter and Author response

Decision letter https://doi.org/10.7554/eLife.75340.sa1
Author response https://doi.org/10.7554/eLife.75340.sa2

## Additional files

### Supplementary files
• MDAR checklist
• Supplementary file 1. Differential expressed genes in Plasma cell cluster 6 (PC6).

### Data availability

Sequencing data have been deposited in GEO under accession code GSE176131.

The following dataset was generated:

| Author(s) | Year | Dataset title | Dataset URL | Database and Identifier |
| --- | --- | --- | --- | --- |
| Zhong L | 2022 | Identification of heterogeneous networks and plasma-monocytes crosstalks in multiple myeloma stratified by R-ISS system through single-cell sequencing | https://www.ncbi.nlm.nih.gov/geo/query/acc.cgi?acc=GSE176131 | NCBI Gene Expression Omnibus, GSE176131 |

The following previously published datasets were used:

| Author(s) | Year | Dataset title | Dataset URL | Database and Identifier |
| --- | --- | --- | --- | --- |
| Chng WJ, Huang GF, Chung TH, Sb NG | 2011 | Clinical and biological implications of MYC activation: a common difference between MGUS and newly diagnosed multiple myeloma | https://www.ncbi.nlm.nih.gov/geo/query/acc.cgi?acc=GSE6477 | NCBI Gene Expression Omnibus, GSE6477 |
| Ledergor G, Weiner A, Zada M, Wang SY | 2018 | Single cell dissection of plasma cell heterogeneity in symptomatic and asymptomatic myeloma | https://www.ncbi.nlm.nih.gov/geo/query/acc.cgi?acc=GSE117156 | NCBI Gene Expression Omnibus, GSE117156 |

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
