## [Editor Report]

The work constructed a single cell transcriptome atlas of bone marrow for R-ISS-staged multiple myeloma patients. It identified novel plasma cell clusters with clinical significance. The finding can help to develop more potential immunotargets for multiple myeloma.

---

## [Decision Letter]

**Decision letter after peer review:**

Thank you for submitting your article "Revised International Staging System (R-ISS) stage-dependent analysis uncovers oncogenes and potential immunotherapeutic targets in multiple myeloma" for consideration by *eLife*. Your article has been reviewed by 3 peer reviewers, one of whom is a member of our Board of Reviewing Editors, and the evaluation has been overseen by YM Dennis Lo as the Senior Editor. The reviewers have opted to remain anonymous.

Essential revisions:

1) The design of the study and analysis of the data and related applications need further clarification.

2) The English writing needs to be revised to make it more readable.

*Reviewer #1 (Recommendations for the authors):*

Most research on MM ScRNA sequencing was carried out based on the CD138 or CD38 sorted PCs. This study provided new insight on this area because it can provide the analysis in MM microenvironment immune cells with the unsorted cells. Though it has its innovation, the study still had its limitation needing improvement.

1. The RISS III patients may have a different cytogenetic abnormality. The PCs with different cytogenetic abnormality may have their own pathogenetic pathways and gene expression signature, which may be inappropriate to mix all the RISS III samples for analysis.

2. The author uses the high expression levels of CD138 to identify plasma cells, and use CNV to identify the malignant plasma cell population. How did the author identify the "high expression of CD138", What is the CD38 expression level on the 3 cluster PCs with strong copy numbers alterations?

3. Which regression out methods did you use in scale function?

4. The author uses CD138+ markers to define plasma cells, but the GSE117156 dataset uses CD38+CD138+ markers to select cells defined as plasma cells, do you this difference may affect the results in this study?

5. The authors did not evaluate the possible existence of double cells in the QC procedure?

6. The author applied criteria to filter out cells with UMI/gene numbers out of the limit of mean value {plus minus} 2 fold of standard deviations assuming a Gaussian distribution of each cells' UMI/gene numbers. Could the author provide the data to show the difference between before and after quality control?

*Reviewer #2 (Recommendations for the authors):*

The current study provides 11 scRNA-seq datasets that should benefit many other researchers on malignant hematology disease. This work is interesting and worth publishing.

The quality of the figure and language should be improved: the authors might polish their language in scientific language editing services.

*Reviewer #3 (Recommendations for the authors):*

1) This study identified a small set of GZMA+ cytotoxic PCs that could be found in public data. This cluster is very small and expresses other cell types maker gene. Could you check if these cells express T cell marker e.g. CD3D? Could you analyze doublets using public tools?

2) The malignant PC population with high proliferation capability was clustered into 3 groups. Did you remove the cell cycle effect before you calculate PCA?

3) MM is plasma cell malignancy. Could you take all B and plasma cells out and re-analyze them? So, you can study heterogeneity well.

4) T cell population was grouped into 22 clusters. Could you annotate these clusters well?

5) Do cell-cell communication with more cell type/clusters, at least more T cell types?

6) You check the copy number for B and plasma. Could you analyze copy number change for all cell types?

7) The writing needs to be significantly improved.

8) The figure resolution is not high enough.

---

## [Author Response]

Essential revisions:1) The design of the study and analysis of the data and related applications need further clarification.

Thanks for your suggestion. We have re-clarified the study design, re-analyzed the data and re-edited the text in the revised version according to your comments.

2) The English writing needs to be revised to make it more readable.

Thanks for your suggestion and comment. We thoroughly polished the language in this version. We hope that our revisions have appropriately addressed your concerns.

Reviewer #1 (Recommendations for the authors):Most research on MM ScRNA sequencing was carried out based on the CD138 or CD38 sorted PCs. This study provided new insight on this area because it can provide the analysis in MM microenvironment immune cells with the unsorted cells. Though it has its innovation, the study still had its limitation needing improvement.1. The RISS III patients may have a different cytogenetic abnormality. The PCs with different cytogenetic abnormality may have their own pathogenetic pathways and gene expression signature, which may be inappropriate to mix all the RISS III samples for analysis.

Thanks for your suggestion and comment. We agreed that the cytogenetic heterogeneities existed in RISS-III MM patients, and compared the cytogenetic heterogeneities with InferCNV. We found that there were discrepancies in different samples, especially MM-III-3 exhibited obvious amplication of chromosome 1 (shown in revised Figure 3—figure supplement 1). Therefore, we have added the description in the main text as ‘The discrepancies existed in the five MM-III samples and cytogenetic CNV signals of sample 4 and 5 are similar to each other. And MM-III-3 exhibited obvious amplication of chromosome 1(Figure. S4)’ accordingly in Line 136-139, Page5

However, at present, there is no article to study the heterogeneities among R-ISS stags at the single-cell level. Therefore, our study focused on the heterogeneities among three stages to reveal the potential mechanism of diverse prognosis.

2. The author uses the high expression levels of CD138 to identify plasma cells, and use CNV to identify the malignant plasma cell population. How did the author identify the "high expression of CD138", What is the CD38 expression level on the 3 cluster PCs with strong copy numbers alterations?

We thank the reviewer for the comments and suggestions. In this study, high expression of CD138 (SDC1) was identified as the positive percentage of CD138 more than 70% in a cluster. Furthermore, there were no other types of cells except plasma cells expressed CD138 in bone marrow. Our result showed that the PCs (PCC4 and PCC6) with strong copy numbers alterations also displayed obvious positive for CD38 expression. The trend was consistent with CD138 (Figure 2—figure supplement 1). Therefore, the identified plasma cells can be labeled as ‘CD138+CD38+’ cells. We have re-edited it in the main text as ‘six clusters (4, 6, 13, 14, 18 and 22) specifically expressing high levels of CD138 (Figure. S2) were classified into plasma cells. The plasma cells also displayed obvious positive for CD38 expression. Therefore, the identified plasma cells can be labeled as ‘CD138+CD38+’ cells in this study.’ in Lines 103-106, Page 4.

3. Which regression out methods did you use in scale function?

We thank the reviewer’s useful comment. We have described the methods in Material and methods section and added them in the main text as ‘In scale function, the regression out method we used was ScaleData (object = seurat_ob, vars.to.regress =c(‘nCount_RNA’, ‘percent.mito’ ))’ accordingly (Lines 713-714, Page 26).

4. The author uses CD138+ markers to define plasma cells, but the GSE117156 dataset uses CD38+CD138+ markers to select cells defined as plasma cells, do you this difference may affect the results in this study?

Thanks for your comment. In bone marrow, CD138 is only expressed in plasma cells. CD38 is not only expressed in the plasma cells, but also expressed in various cells such as T, B cells and myeloid cells. tSNE maps of CD138 and CD38 expression in all cell groups were shown in Figure 2—figure supplement 1. Therefore, CD138 was used to define plasma cell in our study.

5. The authors did not evaluate the possible existence of double cells in the QC procedure?

Thanks for your suggestion, which is very helpful to improve the quality of the manuscript. We agreed that double cells (doublets) could weaken the conclusions for subsequent analysis, such as cell type identification. According to your suggestion, we have removed double cells and re-conducted whole analysis in the revised manuscript. All of the present results (in the revised manuscript) were based on the analysis without double cells.

6. The author applied criteria to filter out cells with UMI/gene numbers out of the limit of mean value {plus minus} 2 fold of standard deviations assuming a Gaussian distribution of each cells' UMI/gene numbers. Could the author provide the data to show the difference between before and after quality control?

Thanks for your comment. We used the criteria to filter out cells according to previous studies [Liu, H., *et al.*, Cell Rep, 2021. 36(11): p.109718.] [Lu, T., *et al.*, Cell Discov, 2020. 6: p.69.]. The UMAP before and after removal of batch-effect (one of the important quality control step) did not have non-biased distribution between samples in stages in revised Figure 1—figure supplement 1A and 1B (Lines 94-95, Page 4).

Reviewer #2 (Recommendations for the authors):The current study provides 11 scRNA-seq datasets that should benefit many other researchers on malignant hematology disease. This work is interesting and worth publishing.The quality of the figure and language should be improved: the authors might polish their language in scientific language editing services.

Thanks for your suggestion and comment. We thoroughly polished the language in this version. We hope that our revisions have appropriately addressed your concerns.

Reviewer #3 (Recommendations for the authors):1) This study identified a small set of GZMA+ cytotoxic PCs that could be found in public data. This cluster is very small and expresses other cell types maker gene. Could you check if these cells express T cell marker e.g. CD3D? Could you analyze doublets using public tools?

Thanks for your suggestion. The expression of T cell marker CD3D/CD3G was provided in Figure 6—figure supplement 1. In this study, we have analyzed the doublets and found that the cytotoxic plasma cells do not express CD3D and other T cell markers shown in revised Figure 6—figure supplement 1.

2) The malignant PC population with high proliferation capability was clustered into 3 groups. Did you remove the cell cycle effect before you calculate PCA?

Thanks for your suggestion. As we focused on the differences in proliferative capacity of myeloma cells, the cell cycle could reflect the difference well. Therefore, the cell cycle data was not removed.

3) MM is plasma cell malignancy. Could you take all B and plasma cells out and re-analyze them? So, you can study heterogeneity well.

Thanks for your suggestion. Several published studies on scRNA-seq in MM have been taken them out to analyze. We intended to explore the crosstalk between plasma cells and microenvironment. Therefore, we didn’t sort them out for analysis.

4) T cell population was grouped into 22 clusters. Could you annotate these clusters well?

We thank the reviewer’s useful and helpful comments and suggestions. The T cell population was annotated with markers, shown in Figure 6—figure supplement 1. According to the expression of T cell markers [Guo X, et al. Nat Med. 2018], the 21 clusters was annotated to naïve (CD4 and CD8) T cells, inhibitory (CD4 and CD8) T cells, effector (CD4 and CD8) T cells and Tregs. The Reference is Global characterization of T cells in non-small-cell lung cancer by single-cell sequencing.

5) Do cell-cell communication with more cell type/clusters, at least more T cell types?

Thanks for your comment. According to your suggestion, we have performed the analysis for the Cell-cell communication with more cell types/clusters of T cell. Based on the number of ligand-receptor pairs, T21 subcluster showed the most cell-cell interactions, and T3 and T15 displayed fewer crosstalk pairs (Author response image 1) for your reference.

**Author response image 1. sa2fig1:** 

6) You check the copy number for B and plasma. Could you analyze copy number change for all cell types?

We thank the reviewer’s comments. MM characterized as malignancies in plasma cells, so we checked the copy number of B and plasma cells with inferCNV algorithm. In the analysis of inferCNV, we assumed and used myeloid cells as control. The results showed patients in stage-III had different pattern of inferCNV, such as amplication of chromosome 1 (Figure 3—figure supplement 1). Information about this description was added in the main text as ‘Abnormal copy number variations (CNV) are a common feature of MM, which usually interferes with cell cycle checkpoints to induce accelerated proliferation[30]. To characterize malignant PCs and related oncogenes in MM, we first conducted InferCNV to delineate the CNV signals of all plasma clusters, especially in the five MM-III samples (Figure. S4). The discrepancies existed in the five MM-III samples. and cytogenetic CNV signals of sample 4 and 5 are similar to each other.’ in Figure 3—figure supplement 1, and Lines 133-138, Page 5.

7) The writing needs to be significantly improved.

Thanks for your suggestion. We thoroughly polished the language in this version. We hope that our revisions have appropriately addressed your concerns.

8) The figure resolution is not high enough.

Thanks for your suggestion. We have improved quality of all the figures and provided the figures in PDF Format with high resolution.